# S235JRC+C Steel Response Analysis Subjected to Uniaxial Stress Tests in the Area of High Temperatures and Material Fatigue

**Josip Brnic** [1,*]⊙, **Marino Brcic** [1]⊙, **Sebastian Balos** [2]⊙, **Goran Vukelic** [3]⊙, **Sanjin Krscanski** [1]⊙, **Mladomir Milutinovic** [2]⊙ and **Miroslav Dramicanin** [2]⊙

[1] Department of Engineering Mechanics, Faculty of Engineering, University of Rijeka, Vukovarska 58, 51000 Rijeka, Croatia; mbrcic@riteh.hr (M.B.); sanjink@riteh.hr (S.K.)

[2] Department of Production Engineering, Faculty of Technical Sciences, University of Novi Sad, Trg Dositeja Obradovića 6, 21000 Novi Sad, Serbia; sebab@uns.ac.rs (S.B.); mladomil@uns.ac.rs (M.M.); dramicanin@uns.ac.rs (M.D.)

[3] Faculty of Maritime Studies, University of Rijeka, Studentska ulica 2, 51000 Rijeka, Croatia; gvukelic@pfri.hr

[*] Correspondence: brnic@riteh.hr; Tel.: +385-51-651-491

**Abstract:** Knowledge of the properties and behavior of materials under certain working conditions is the basis for the selection of the proper material for the design of a new structure. This paper deals with experimental investigations of the mechanical properties of unalloyed high quality steel S235JRC + C (1.0122) and its behavior under conditions of high temperatures, creep and mechanical fatigue. The response of the material at high temperatures (20–700 °C) is shown in the form of engineering stress-strain diagrams while that at creep behavior (400–600 °C) is shown in the form of creep curves. Furthermore, based on uniaxial fully reversed mechanical fatigue tests ($R = -1$), a stress-life (S-N) fatigue diagram has been constructed and the fatigue (endurance) limit of the material is calculated The experimentally determined value of tensile strength at room temperature is 534 MPa. The calculated value of the fatigue limit, also at room temperature, using the modified staircase method and based on the mechanical fatigue tests data, is 202 MPa. With regard to creep resistance, steel 1.0122 can be considered creep-resistant only at a temperature of 400 °C and at an applied stress not exceeding 50% of the yield strength corresponding to this temperature.

**Keywords:** mechanical properties; uniaxial creep; uniaxial fatigue; fatigue limit; structural steel 1.0122

## 1. Introduction

The choice of material from which an element will be made is related to the purpose of the element but also depends on the process by which the element will be shaped and joined [1]. The finite element method [2] is usually considered an appropriate technique in the analysis of the mechanical behavior of a structure, where the mechanical behavior of the structure is understood as the study of its deformation and fracture [3]. Certain knowledge about mechanical failure modes causing deformation can serve as the basis for avoiding excess deformation leading to complete fracture of the engineering element. To the group of the mentioned and often observed modes of the mechanical failures belong yielding, impact, creep, fatigue, corrosion, etc. It is possible to distinguish two basic groups of failures, namely, pre-existing material failures and failures that can be created during structure life. In this sense, creep means deformation that accumulates with time and it is thermally activated phenomenon, fatigue is failure due to repeated loading and causes fracture, etc., [4,5]. In terms of creep, this phenomenon is commonly defined as a time-dependent inelastic strain under sustained load and elevated temperature [6]. For metallic materials, it is most often represented by a curve having three distinct areas such as the primary, secondary and tertiary creep stage. The first (primary, transient) creep

stage in creep process represents creep region with decreasing creep rate where the creep resistance of material is increased by its own deformation. The second (steady-state) creep stage exhibits nearly constant creep rate and is called region of minimum creep rate which results from balance processes between strain hardening and recovery. Finally, the third (tertiary, accelerating) creep stage, often associated with metallurgical changes (such as diffusional or recrystallization changes, coarsening of precipitates) and results in necking (reduction in cross-sectional area) of the element. An amount of (1–2)% creep strains in engineering practice is allowed, and creep is appreciable above 0.4 of melting temperature [7]. Vacancy diffusion, dislocation climb and grain boundary sliding are usually numbered as mechanisms of creep.

Except of considering uniaxial creep causing axial extension, under uniaxial compression creep occurs not only in axial direction but also in lateral direction. This appearance is referred to as lateral creep. In analogy with elastic strains, the ratio between lateral creep and axial creep is termed creep Poisson's ratio. Under axial compression, normal (lateral) creep is an extension. The tendency of the material to creep can be manifested at different types of loads. Usually, tensile creep is preferred test method for rupture, since some materials cannot rupture, for example, in compression or flexural mode. Since S235JRC + C steel is cold worked, this indicates an improvement in its yield strength. Experimental research, conducted in this paper, shows its weak creep resistance. The experiments were made to see how much such resistance really exists, since the material can get into a hazardous situation. In engineering practice, operating of the structural element under tertiary creep conditions is not allowed. In this study, experimental tests of uniaxial creep were performed at temperatures of 400 °C, 500 °C and 600 °C, where the level of applied stress at a certain temperature was defined as a percentage of the yield strength of the material corresponding to the creep temperature. Creep curves are shown later.

With regard to fatigue it was found experimentally that the load-bearing capacity of the material during repeated loading is significantly reduced compared to the load bearing capacity of the material at monotonic load. The main purpose of this research is to determine the changes in the mechanical properties of non-alloy structural steel S235JRC + C with a change in temperature as well as its behavior in creep and fatigue regimes. It is not possible to find similar consolidated content of this material in the literature. Such research provides structure designers with an insight into the possibilities of applying this material in the specific operating conditions of the structure. In the following part, several published papers related to the considered material are mentioned, as well as several published papers related to other materials exposed to similar loads, in order to compare their behavior. In that sense, the published paper [8] deals with determination of the residual stress distribution of tungsten inert gas welded S235JRC + C plates. In [9], the critical monotonic strain of Ni-W and $MoS_2$ coatings on steel substrate was studied. An axisymmetric bending test limited to monitoring of the coating failure was used. The specimens were disc-shaped coated on one side. The discs were made of S235JRC cold drawn steel and K340 steel manufactured by Bohler. In [10] the fatigue response of electrodeposited Ni-W on low carbon cold drawn steel (S235JRC) discs was studied, and it was considered as promising substitute for toxic hard chromium coatings. Further, in [11] the change of the ultimate tensile strength and the change of yield strength related to hot rolled S235JR steel and cold drawn S235JRC steel was investigated and analyzed. These test results were needed to compare the mechanical properties of two grades of steel, but also as a starting point for determining the change in mechanical properties during the development of corrosion. Similarly, in [12] an engineering stress-strain diagram of S235JRC steel is shown. With regard to possible application of S235JRC steel, in [13] the rupture of a galvanized U-bolt steel stirrup of a 60 kV overhead electric transport line was reported. The mentioned component was subjected to a complex load system and variable attack angles. Thus, the component was subjected to fatigue wear, static crush and corrosion. Further, in [14] the authors have investigated the microstructure, residual stress as well as magnetic anisotropy of the ship structure elements made of S235 steel after uniaxial tensile deformation.

## 2. Data Relevant for Research

*Material*

The material under consideration was unalloyed high quality structural cold drawn round (18 mm) steel bar S235JRC + C (1.0122). It is the most frequently used steel grade worldwide. Due to the fact that the percentage of sulfur and phosphorus content is less than 0.4% this material can be assigned to the group of so called "quality steels". Its chemical composition is shown in Table 1. The chemical composition of the alloy was determined using a GDS500A optical emission spectrometer (LECO, St Joseph, MI, USA) by the method of oscillating discharge (argon 99.999%), as stated in the report of the institution ("Metris"-Pula) certified for this activity. This steel is classified as steel for general engineering purposes ("structural steel"). Regarding the applications of steel 1.0122 in engineering areas such as civil engineering, mechanical engineering, bridge building, vehicle industry, etc., can be mentioned. The applications are mainly related to low loaded components. It is good for machining and welding. Although it does not belong to the group of materials that are resistant to certain environmental conditions, due to its wide use it can be found in various, even unfavorable, environmental conditions such as elevated temperature or the like. Due to the favorable level of ultimate tensile strength and yield strength, it is often termed as "steel for the construction industry". It is also classified as a mild (low-carbon, plain) steel.

**Table 1.** Chemical composition of the material.

| Tested Material Mass (%) | | | | | | | |
|---|---|---|---|---|---|---|---|
| C | Si | Mn | P | S | Cr | Ni | Mo |
| 0.162 | 0.237 | 0.534 | 0.011 | 0.012 | 0.158 | 0.028 | 0.009 |
| Cu | Al | W | Sn | Nb | Co | Rest | |
| 0.039 | 0.027 | 0.003 | 0.008 | 0.003 | 0.002 | 98.767 | |

Classification: Unalloyed high-quality structural steel/steel for construction industry
Chemical composition (Certified Labor. "Metris", Pula)

The equipment used to determine the mechanical properties of the material and to test the creep behavior was a 400 kN material testing machine (Zwick/Roell, Ulm, Germany), while in fatigue testing a Hydropuls PSB, 250 kN hydraulic pulsator was used Schenck (Aachen, Germany). In addition, in high temperature tests, a high temperature furnace (900 °C) as well as a high temperature extensometer, both Maytech (Singen, Germany) products, were used. Microstructure of the material was analyzed using a scanning electron microscope (SEM, JSM6460LV, JEOL, Tokyo, Japan). Uniaxial tests were performed to determine the mechanical properties, creep behavior and fatigue of the material.

Test specimens used in the tests are of different geometries depending on the type of tests, Figure 1. All of specimens used were smooth highly polished specimens.

The standards that define the particular test procedure as well as both the manufacture and the geometry of test specimens are as follows. ASTM: E8M-16a (2015) standard defines the geometry of a test specimen used in uniaxial creep tests as well as in uniaxial tests to determine the mechanical properties (room and high temperatures). ASTM: E21-17e1 (2015) standard defines the test procedure related to uniaxial tests for determination of mechanical properties at high temperatures, while ASTM: E139-11 (2018) standard defines the test procedure related to creep testing. In addition, fatigue testing procedure and the geometry of specimen used in fatigue tests are defined by ISO 12107 (2012) standard [15]. All of mentioned ASTM standards can also be found in [16].

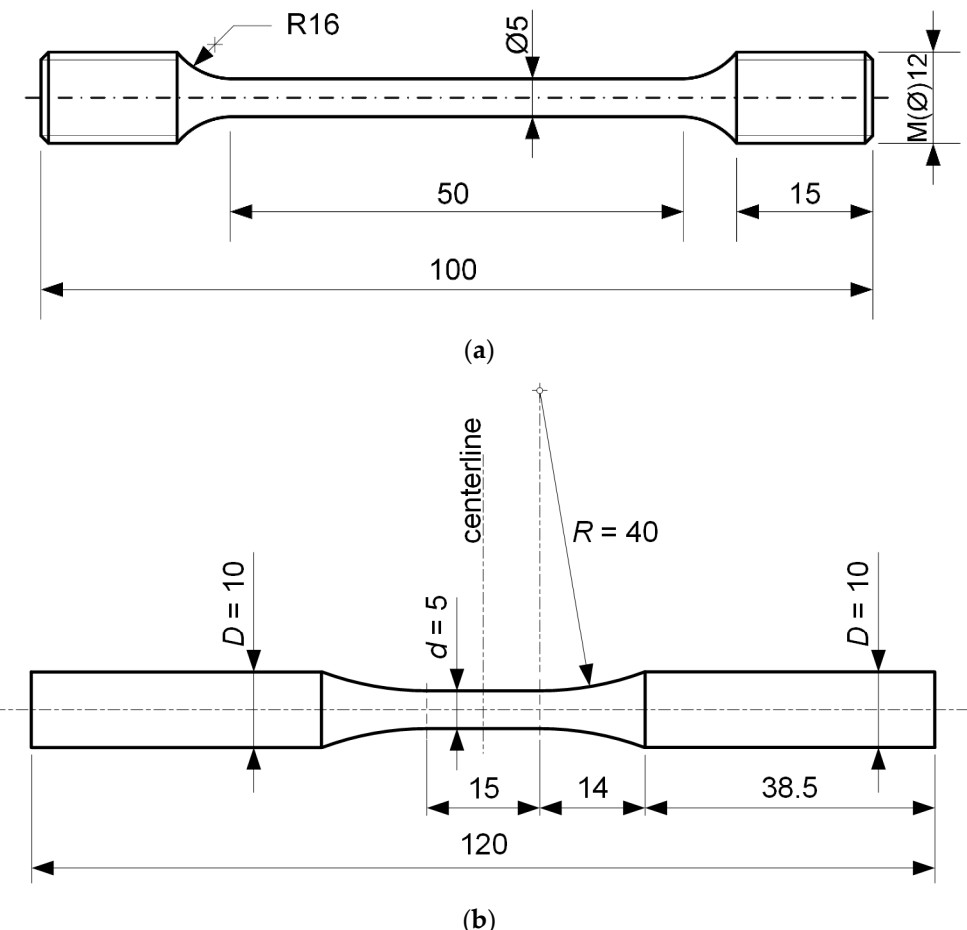

**Figure 1.** Specimens used in research (mm). (**a**) Specimen used in determination of engineering stress-strain diagrams and creep curves. (**b**) Specimen used in fatigue tests.

## 3. Results and Discussion

### 3.1. Tensile Engineering Stress-Strain Diagrams and Mechanical Properties Versus Temperature

All tests related to the determination of the mechanical properties of the material were performed using a computer-controlled machine based on the mentioned standards in terms of sample geometry, test procedure and test parameters. The increase in temperature during testing leads to a change in the shape of the engineering stress-strain diagrams of the material under consideration, and in this sense indicates a change in the mechanical properties of the material. Several tests were performed at some selected temperatures, and, it was found that the obtained diagrams for the same temperature differ very slightly from each other. For that reason duplicate diagrams were rejected and all shown diagrams refer to the first test at each tested temperature (Figure 2a,b). In addition, in Table 2 numerical values of mechanical properties versus temperature are presented. Figure 2c–e show changes in properties as a set of discrete experimental data.

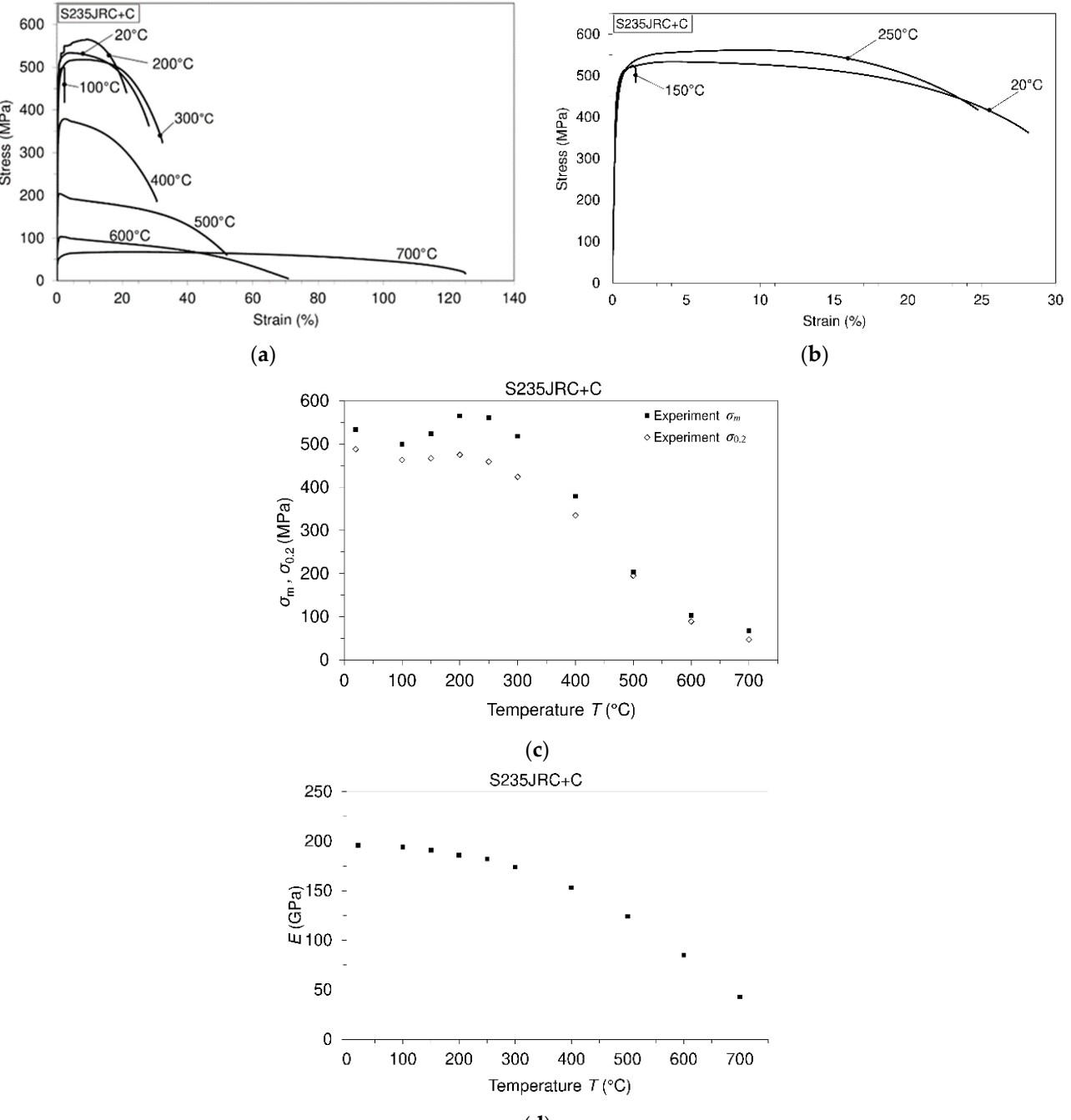

**Figure 2.** *Cont.*

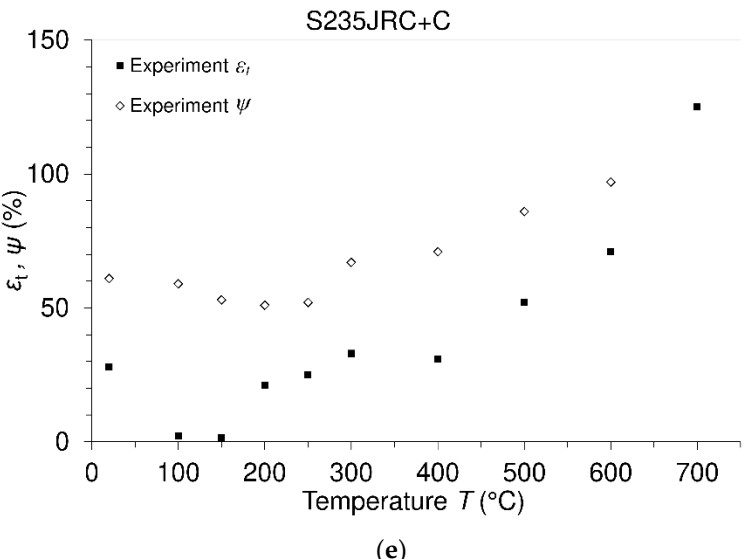

(**e**)

**Figure 2.** Engineering stress-strain diagrams and mechanical properties versus temperature (**a**) Diagrams related to the temperature: 20 °C, 100 °C, 200 °C, 300 °C, 400 °C, 500 °C, 600 °C, 700 °C. (**b**) Diagrams related to the temperature: 20 °C, 150 °C, 250 °C. (**c**) Ultimate tensile strength ($\sigma_m$) and yield strength ($\sigma_{0.2}$) versus temperature. (**d**) Modulus of elasticity ($E$) versus temperature. (**e**) Total strain ($\varepsilon_t$) and reduction of cross-sectional area ($\psi$) versus temperature (related to the specimen).

**Table 2.** Numerical values of mechanical properties versus temperature.

| Temp. $T$ (°C) | Ultimate Tensile Strength ($\sigma_m$), Yield Strength ($\sigma_{0.2}$) and Modulus of Elasticity ($E$) | | | | Reduction Factor ($f$) $f = F_{i,T}/F_{i,20}$; $F_i = \sigma_m, \sigma_{0.2}, E$ | | | Total Strain ($\varepsilon_t$) and Reduction in Area ($\psi$)/Specimen/ | |
|---|---|---|---|---|---|---|---|---|---|
| | $\sigma_m$ (MPa) | $\sigma_{0.2}$ (MPa) | Ratio $\sigma_{0.2}/\sigma_m$ | $E$ (GPa) | $\sigma_{m,T}/\sigma_{m,20}$ | $\sigma_{0.2,T}/\sigma_{0.2,20}$ | $E_{,T}/E_{,20}$ | $\varepsilon_t$ (%) | $\psi$ (%) |
| 20 | 534 | 488 | 0.93 | 196 | 1 | 1 | 1 | 28 | 61 |
| 100 | 500 | 463 | 0.93 | 194 | 0.94 | 0.95 | 0.99 | 2.3 | 59 |
| 150 | 524 | 467 | 0.89 | 191 | 0.94 | 0.95 | 0.89 | 1.6 | 53 |
| 200 | 565 | 475 | 0.84 | 186 | 1.01 | 0.96 | 0.66 | 21 | 51 |
| 250 | 561 | 459 | 082 | 182 | 1.007 | 0.93 | 0.87 | 25 | 52 |
| 300 | 518 | 424 | 0.82 | 174 | 0.93 | 0.86 | 0.81 | 33 | 67 |
| 400 | 379 | 335 | 0.88 | 153 | 0.68 | 0.68 | 0.819 | 31 | 71 |
| 500 | 204 | 195 | 0.94 | 124 | 0.481 | 0.39 | 0.71 | 52 | 86 |
| 600 | 103 | 89 | 0.96 | 85 | 0.37 | 0.29 | 0.4 | 71 | 97 |
| 700 | 67 | 47 | 0.7 | 43 | 0.12 | 0.1 | 0.2 | 125 | ---- |

Based on the experimental tensile tests (Figure 2a,b), as well as the data given in Table 2, a trend of change of mechanical properties with increasing temperature is visible. Tensile strength and yield strength of the material decrease with increasing temperature from room temperature to a temperature of 100 °C. After this temperature, both properties increase with increasing temperature and the tensile strength reaches a maximum of 565 MPa at a temperature of 200 °C, while yield strength, at this temperature, reaches a relative maximum of 475 MPa. After that, both properties decrease with increasing temperature (Figure 2c). The value of the modulus of elasticity continuously decreases with temperature increase (Figure 2d). Based on this data, it is also evident that the material has sufficiently high level of mechanical properties up to a temperature of 400 °C, which is advantageous given the use of this material. On the other hand, Figure 2a,b and Figure 2e, considering the elongation, i.e., the change in deformation, a certain phenomenon appears in the temperature area of 100 °C, 150 °C and 200 °C. Namely, at these temperatures the

strains of the material decrease, i.e., they are somewhat reduced in comparison with those at higher temperatures. However, this appearance is a consequence of dynamic strain aging phenomenon which is treated as hardening phenomenon. Usually, as the most visible effect of dynamic strain aging is the appearance of serrations in the stress-strain curve (known as the Portevin-Le Chatelier effect). When the serrations are not seen, this phenomenon can be marked by lower strain rate sensitivity and also can cause variation in ductility with temperature or, for example, a plateau in strength. In terms of research into the properties and mechanical behavior of S235JRC material, the number of papers in the literature is quite limited. The chemical composition of the material as well as the experimentally obtained data on the mechanical properties of S235JRC + C steel in this study, were compared with those provided by the standard and with the data of another study (Table 3).

**Table 3.** Data comparison: Chemical composition and mechanical properties-standard, this research and Ref. [11].

| Chemical Composition (%) | C | Si | Mn | P | S | Cr | Cu | Properties | | |
|---|---|---|---|---|---|---|---|---|---|---|
| | | | | | | | | $\sigma_m$ (MPa) | $\sigma_{0.2}$ (MPa) | $E$ (GPa) |
| EN 10277-2-2008; S235JRC/1.0122; (max) | 0.17 | - | 1.4 | 0.04 | 0.04 | - | 0.55 | Specimen: 16–40 mm, (+C) 390–730 | ≥260 | - |
| S235JRC + C (This paper)/cold drawn | 0.162 | 0.237 | 0.534 | 0.011 | 0.012 | 0.158 | 0.039 | 534 | 488 | 196 |
| S235JRC, Ref. [11])/cold drawn | 0.08 | 0.2 | 0.08 | 0.027 | 0.027 | 0.1 | 0.55 | 609 | 559 | 208 |

By comparing the data given in the Table 3 it is possible to determine the following:

- the chemical composition of the test material specified in this paper as well as chemical composition of the material specified in [11] is within the limits set by the EN standard.
- the values of the tested mechanical properties of steel S235JRC + C shown in this paper and those shown for the material S235JRC in [11], differ from each other.

The values of ultimate tensile strength, yield strength and modulus of elasticity of S235JRC shown in [11] are higher by 14%, 14.5% and 6%, respectively, compared to those related to the S235JRC + C steel tested in this study (this paper). Furthermore, the ultimate tensile strength values of S235JRC + C steel obtained at high temperatures in this study, can be compared with those relating to some other low-carbon (plain) steels such as AISI 1008 and AISI 1020, shown in [17]. It was found experimentally that the highest value of ultimate tensile strength for each of the three compared low carbon (plain) steels (S235JRC + C, AISI 1008 and AISI 1020) occurs at a temperature of about 200 °C. The data are shown in Table 4. Dynamic strain aging is considered to be the cause of increased flow stress with an increase in temperature up to about 200 °C.

**Table 4.** Data comparison: Low-carbon (plain) steels—Ultimate tensile strength at temperature of $T = 200$ °C

| Low-Carbon (Plain) Steels | AISI 1008 (1.0204) Ref. [17] | AISI 1020 (1.0402) Ref. [17] | S235JRC + C (1.0122) (This Study) |
|---|---|---|---|
| Ultimate tensile strength at $T = 200$ °C (considered as maximum) MPa;/$\sigma_{m,200°C} = \sigma_{m,\,max}$/ | 400 | 550 | 565 |

However in order to select the appropriate material for the purpose of a particular application, it is of interest to have insight into the behavior of some materials at prescribed environmental conditions. In this regard, in [18] the mechanical properties of a martensitic steel were investigated.

*3.2. Tensile Short Time Creep Tests*

Since S235JRC + C steel is cold worked, this indicates an improvement in its yield strength. Experimental research, conducted in this paper, shows its weak creep resistance. The experiments were made to see how much such resistance really exists, since the material can get into a hazardous situation. In engineering practice, operating of the structural element under tertiary creep conditions is not allowed. In this study, experimental tests of uniaxial creep were performed at temperatures of 400 °C, 500 °C and 600 °C, where the level of applied stress at a certain temperature was defined as a percentage of the yield strength of the material corresponding to the creep temperature. In Figure 3 creep curves are shown. All of creep curves shown in Figure 3 were obtain using material testing machine Zwick/Roell and Maytech furnace (max 900 °C), while deformation (strain) during the time is measured by high temperature extensometer. Experimental procedure, related to any test temperature and any applied stress level takes place as follows. First, in accordance with the standard, specimen is mounted in the furnace which has three thermos- couple. the creep strain is monitored and recorded, i.e., the creep curve is plotted.

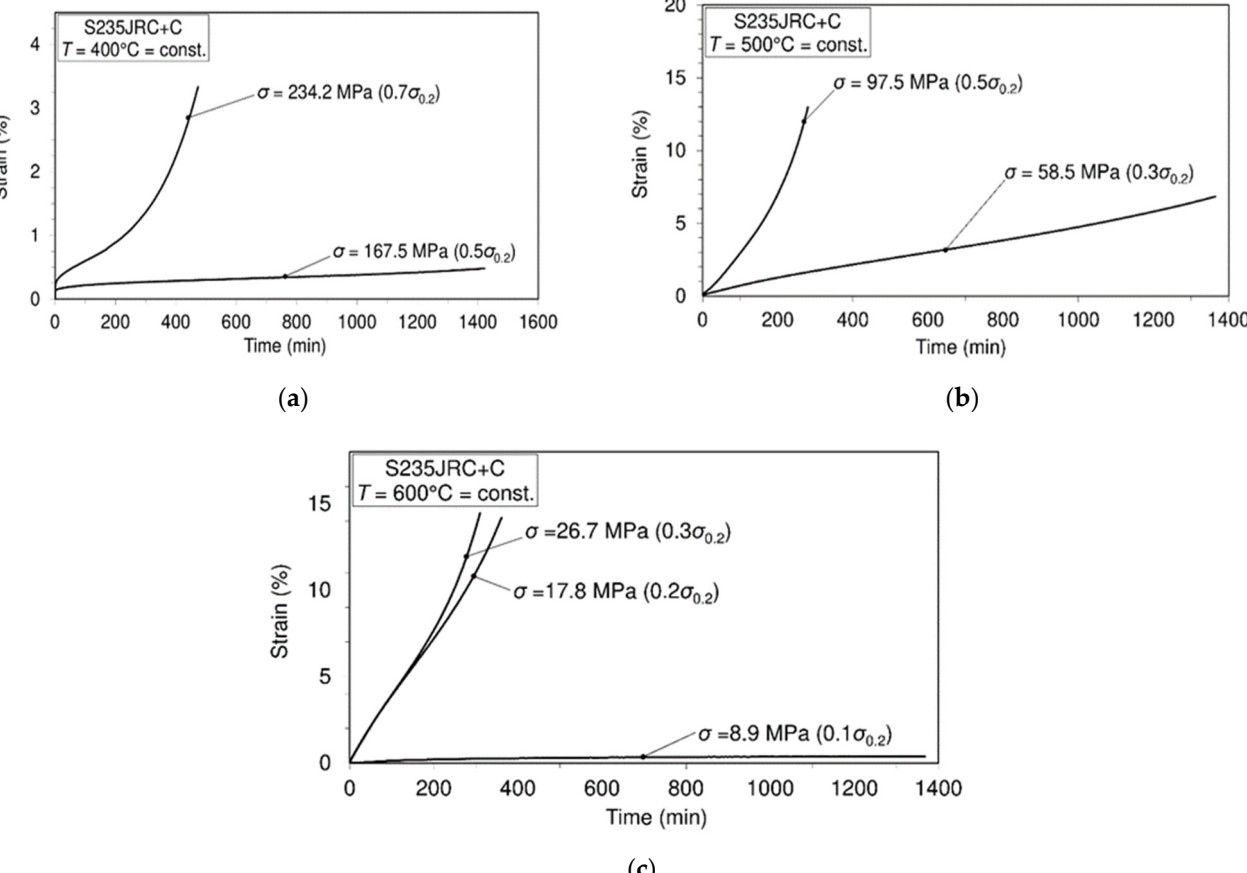

**Figure 3.** Creep curves, at: (**a**) 400 °C. (**b**) 500 °C. (**c**) 600 °C.

The temperature is adjusted at the desired level and the specimen is continuously and completely heated at the test temperature level. Using testing machine a minimum load is applied, approximately (10–15) N, which is kept constant for 60 min and during this time specimen is evenly and completely heated. This low level applied stress allows the specimen to be kept to a minimum stress. After this time, since testing machine is computer controlled and directed, further, next part of the test program is activated in such a way that temperature is still kept constant but stress (load) is increased to the proposed test stress level. This stress level is now also kept constant during the test time, that is, the duration of test, while.

This material shows a creep resistance only at a temperature of 400 °C and in the case when the applied stress does not exceed 50% of the yield strength of the material at that temperature. At higher temperatures (500 °C, 600 °C) it is evident that this material has no creep resistance even at low applied stresses. Namely, the deformation then crosses the limit allowed in engineering. This was only the reason to show how the material behaves at higher temperatures. On the other hand, different materials exhibit their behavior differently at elevated temperatures. For example, in low-alloy steels elevated temperature behavior occurs at approximately 370 °C. In this sense, testing such behavior at temperatures below 400 °C would have no reason.

Since there is sometimes not enough data on the creep of the considered material in the literature, it is useful, for comparison, to have an insight into the creep behavior of other material [19].

### 3.3. Uniaxial Fully Reversed Mechanical High-Cyclic Fatigue Tests Performed on Unnotched Specimens

Engineering practice points to the fact that in cases when metals are subjected to a fluctuating (dynamic, repetitive) load, the failure occurs at a stress level that is significantly lower than the fracture stress corresponding to a monotonic load of the same type. The reason for this is material fatigue, which is one of the mechanical failures. Namely, due to the fatigue of the material during the operation of the engineering element (structure), a crack occurs in the element and crack propagation ultimately leads to the fracture of the element. In accordance with the above, it is necessary to have data on the resistance of the material to fatigue to which the material will be exposed during working conditions. In order to obtain the necessary data, the material is tested for fatigue under conditions that will correspond to the conditions of its exploitation. In this sense a material ability to withstand cyclic fatigue loading can be determined, that is, a material is properly selected that it can meet service fatigue load. Cyclic (repeated) fatigue loading and unloading can occur in several forms such as tension, compression, torsion, bending or combination of these loadings (stresses). There are many different forms of fatigue failures. In this research the so called mechanical fatigue [20] of the material was considered. Fluctuations in externally applied stresses (or strains) result in mechanical fatigue. Other forms of fatigue failures can also be mentioned such as creep-fatigue failure, corrosion failure, etc. However, fatigue testing provides the answer about the lifespan of the material that can be expected in such cyclic loading. Two types of fatigue testing or two forms are commonly mentioned and they are: stress controlled high cycle and strain controlled low cycle fatigue. The first type of tests tends to be associated with stresses (loads) belonging to the elastic regime while another one generally involve plastic deformations. Research whose results are presented in this paper relate to the uniaxial fully reversed mechanical high-cyclic fatigue tests performed on unnotched specimens at room temperature in air atmosphere.

### 3.3.1. Stress-Life (S-N) Diagram

The results of uniaxial fully reversed ($R = -1$) mechanical high-cyclic fatigue tests (sinusoidal shape stress cycle) performed at room temperature on smooth (unnotched) specimens, which led to the fatigue failure (fracture), were entered into the coordinate system. The ordinate of the coordinate system represents the maximum applied stress while the abscissa represents the number of the cycles to failure. In this way, each test in the coordinate system was recorded as a single point in such a way that the test that led to the fatigue failure (fracture) was marked with a sign ($\blacklozenge$), while the one that did not lead to a fatigue failure (fracture), i.e., the specimen remained unfailed, was marked with a sign ($\bigcirc$). The test that did not lead to a fatigue failure (fracture) is considered the test in which the specimen has withstood more than 10 million cycles (specimen remained unbroken after $10^7$ cycles), which is considered common for steel alloys. Fatigue testing that belongs to the stress-life model was carried out in accordance with ISO standard [15] and in a decreasing stress regime. Using such testing procedure, and after the fatigue (endurance) limit ($\sigma_f$) by modified staircase method was calculated (determined), the so called $S - N$ diagram

(stress-life, fatigue-life, Wohler curve) was constructed, Figure 4. Such a diagram, which shows the fatigue behavior of the material, consists of two regions, namely, the area of finite fatigue (life) and the area of infinite fatigue (life). When the diagram is plotted in a linear form then the inclined line represents the finite fatigue region while the horizontal line represents the infinite fatigue region and denotes fatigue limit.

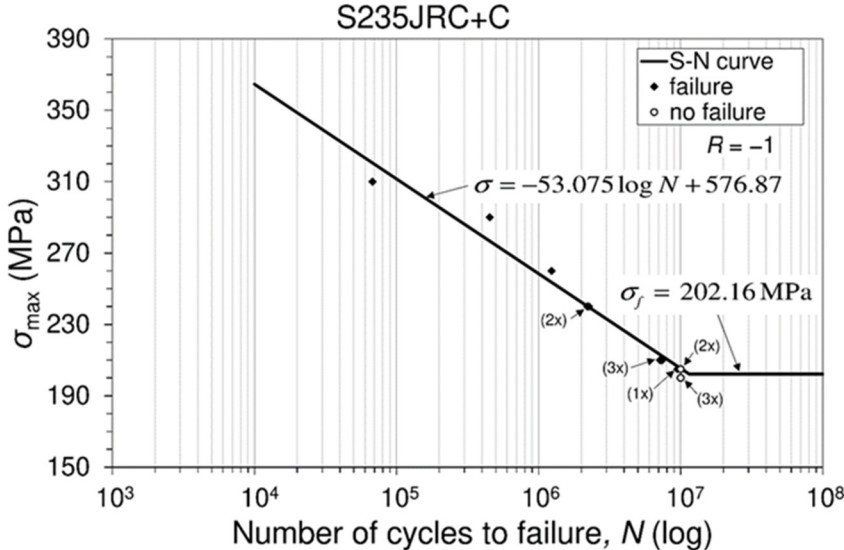

**Figure 4.** Material fatigue test results and associated stress-life $(S - N)$ diagram, stress ratio $R = -1$.

### 3.3.2. Fatigue Limit Calculation

The fatigue (endurance) limit of the tested material was calculated based on the fatigue testing data using modified staircase method. In Table 5, data related to the failed (♦) and unfailed (○) specimens, as well as the applied stress levels that follow from fatigue tests, are shown.

**Table 5.** Data used in the modified staircase method.

| Stress $\sigma_i$/max (MPa) | Stress Ratio $R = -1$, Room Temp., Failed (♦), Unfailed (○) | | | | | | |
|---|---|---|---|---|---|---|---|
| | Specimen | | | | | | |
| | 1 | 2 | 3 | 4 | 5 | 6 | 7 |
| 210 | | | ♦ | | ♦ | | ♦ |
| 205 | | ○ | | ♦ | | ○ | |
| 200 | ○ | | | | | | |

The data given in Table 5 are analyzed as presented in Table 6, while determination of the constants A, B, C and D is displayed in Table 7.

**Table 6.** Data analysis related to Table 4.

| Stress $\sigma_i$/MPa | Stress level, $i$ | $f_i$ | $if_i$ | $i^2 f_i$ |
|---|---|---|---|---|
| 210 | 2 | 3 | 6 | 12 |
| 205 | 1 | 1 | 1 | 1 |
| 200 | 0 | 0 | 0 | 0 |
| $\sum f_i, if_i, i^2 f_i$ | | 4 | 7 | 13 |

**Table 7.** The constants A, B, C and D, calculated accordingly to ISO 12107 [15].

| Stress Ratio $R = -1$ | |
| --- | --- |
| Formula | Tested material: |
| $A = \sum i \times f_i$ | 7 |
| $B = \sum i^2 \times f_i$ | 13 |
| $C = \sum f_i$ | 4 |
| $D = \frac{B \times C - A^2}{C^2}$ | 0.1875 |

The procedure for determining the fatigue limit ($\sigma_f$) according to the ISO standard [15], proceeds as follows:

$$\sigma_{f(P,\,1-\alpha)} = \overline{\mu}_y - k_{(P,1-\alpha,v)} \times \overline{\sigma}_y, \tag{1}$$

The mean fatigue strength ($\overline{\mu}_y$), shown in Equation (1), was calculated as:

$$\overline{\mu}_y = \sigma_0 + d\left(\frac{A}{C} - \frac{1}{2}\right) \tag{2}$$

In Equation (2), $\sigma_0$ is the lowest stress level and "d" is the stress step (the difference between the neighboring stress levels), see Table 6.

To determine the fatigue limit ($\sigma_f$), according to Equation (1), two parameters need to be defined previously, and that:

- $k_{(P,1-\alpha,v)}$, the coefficient for the one sided tolerance limit for a normal distribution, and
- $\overline{\sigma}_y$, the estimated standard deviation of the fatigue strength that can be calculated as:

$$\overline{\sigma}_y = 1.62 \times d(D + 0.029). \tag{3}$$

In accordance with the recommendation of the mentioned ISO standard, the value $v = n - 1 = 6$, where $n$ is the number of items in a considered group. In addition, if a desired probability is $P = 10\%$, and a confidence level $(1 - \alpha) = 90\%$, according to the table B1, given in ISO standard (2012), Ref. [15], and it is: $k_{(P,1-\alpha,v)} = k_{(0.1;0.9;6)} = 2.333$. Finally, according to Equation (2), it is:

- for $R = -1 \to \overline{\mu}_y = \sigma_0 + d\left(\frac{A}{C} - \frac{1}{2}\right)$ = 200 + 5 × (7/4 − 1/2) = 206.25 MPa, or, this item can be obtained as (Table 4):
- for $R = -1 \to \overline{\mu}_y$ = (200 + 205 + 210 + 205 + 210 + 205 + 210)/7 = 206.43 MPa, whose amount is similar to previously obtained one. Now, based on Equation (3), it is:
- for $R = -1 \to \overline{\sigma}_y$ = 1.62 × d(D + 0.029) = 1.62 × 5 × (0.1875 + 0.029) = 1.754 MPa. Finally, fatigue limit is (Equation (1)):
- for $R = -1 \to \sigma_{f(0.1;0.9;6)} = \overline{\mu}_y - k_{(P,1-\alpha,v)} \times \overline{\sigma}_y$ = 206.25 − 2.333 × 1.754 = 202.16 MPa.

Calculated value of the fatigue limit, based on the fatigue testing at stress ratio of $R = 1$, shows that it reaches a level of 38% (=202.16/534) of the ultimate monotonic tensile strength. Although this steel is widely used in engineering practice, it is not easy to find analyzes related to its behaviors in particular situations. In this sense, in [21], an analysis of fatigue prediction of S235 base steel plates in the riveted connections.

### 3.4. A Brief Review of the Microstructure Analysis of the Tested Material

Before reviewing the microstructure of the material investigated in this study, it seems advisable to draw attention to the relationship between the choice of material, its properties and its microstructure. The mechanical properties of steels are closely related to the chemical composition, processing path as well as the microstructure of the considered material. Processing, as a means for developing and control the microstructure may be related to the processes such as hot rolling, cold rolling, quenching, etc. Mechanical properties that depend on the resulting microstructure (yield strength, hardness, etc.), are called structural

sensitive properties. The exposure of the material to certain environmental conditions can be manifested in possible changes in the microstructure of the material. In this regard, the analysis of the microstructure of the considered material is based on the research of the microstructure of three specimens tested in mutually different applications, i.e., conditions. Those specimens are: specimen (1) representing as-received material, specimen (2) representing material previously subjected to creep and specimen (3) representing material previously subjected to fatigue. The analysis of the microstructure of a material characterized by a certain state of it was preceded by metallographic preparation of the surface (cross-sectional and longitudinal) of an individual specimen when it comes to the first two mentioned states of material, ie material as-received (specimen 1) and material previously subjected to creep (specimen 2). In addition, the similar refers also to the fracture surfaces, relating to the specimen 3, to determine microstructure and characterize the surface morphology of fracture. Metallographic preparation of the specimen (sample) surfaces included cutting, grinding, and polishing. Surface etching was performed using nital (2% nitric acid-HNO3 in alcohol). Analysis of the material microstructure for all three mentioned material states (as -received material, material previously subjected to creep and material previously subjected to fatigue) was made by scanning electron microscope (SEM).

Figures related to microstructure analysis are shown as follows. In Figure 5 SEM micrographs related to as-received material are shown.

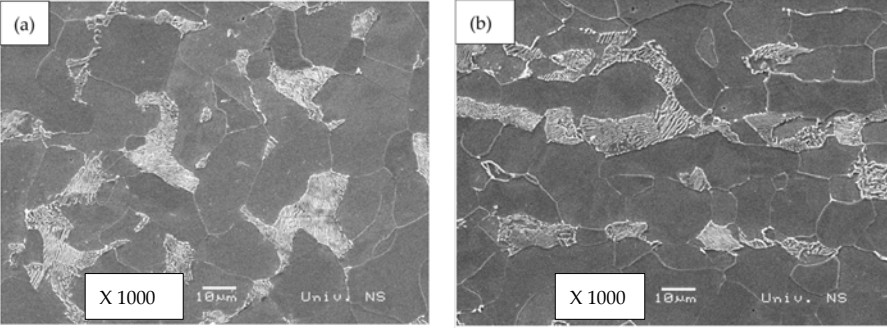

**Figure 5.** SEM micrographs, steel S235JRC + C: Specimen 1 (as –received material). (**a**) Cross-section. (**b**) Longitudinal section

In Figure 6, SEM micrographs related to the material previously subjected to creep are shown.

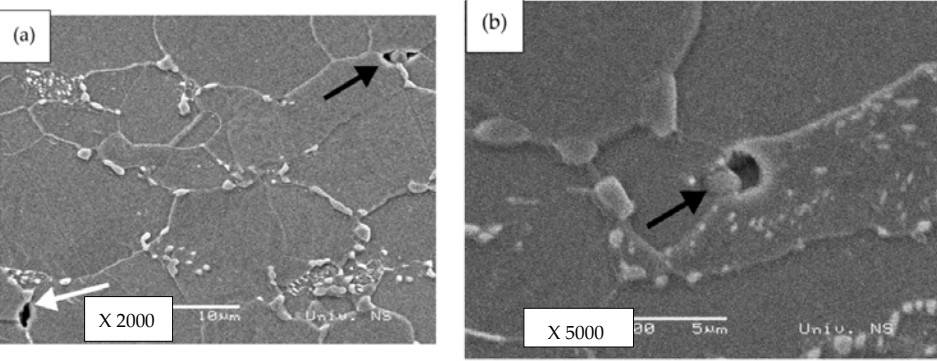

**Figure 6.** SEM micrographs, steel S235JRC + C: Specimen 2 (material previously subjected to creep: *T* = 600 °C max applied stress = 26.7 MPa = 0.3 yield stress at 600 °C,), cross-section. (**a**) Voids occurring at grain boundaries (white arrow) and non-metallic inclusions (black arrow). (**b**) A void that occurred at non-metallic inclusion (higher magnification).

In Figure 7 SEM micrographs related to the material previously subjected to uniaxial fully reversed mechanical high-cyclic fatigue are shown.

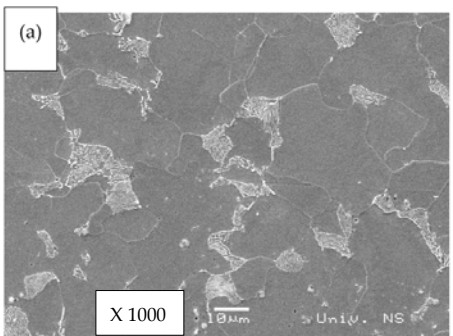 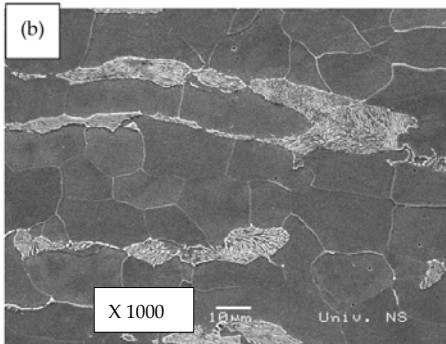

**Figure 7.** SEM micrographs, steel S235JRC + C: Specimen 3 (material previously subjected to uniaxial fully reversed mechanical fatigue, room temperature in air atmosphere; specimen fractured after 7,142,296 cycles under max applied stress of ±210 MPa, 25 Hz). (**a**) Cross-section. (**b**) Longitudinal section.

In addition, Figure 8. shows the fracture surface of a failed (fractured) specimen (specimen 3) due to uniaxial fully reversed mechanical high-cyclic fatigue testing.

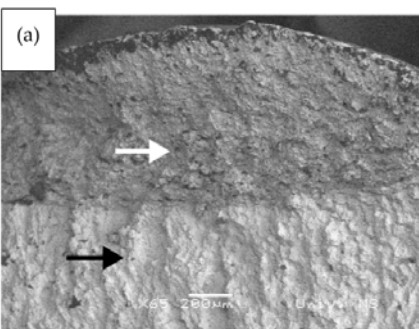

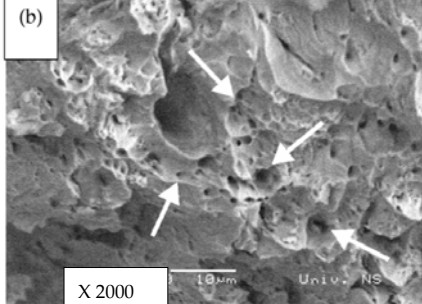 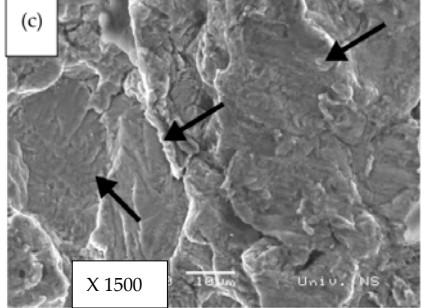

**Figure 8.** The specimen fracture surface due to fatigue, steel S235JRC + C: Specimen 3 (uniaxial fully reversed mechanical fatigue, room temperature in air atmosphere, specimen fractured after 7,142,296 cycles under max applied stress of ±210 MPa, 25 Hz). (**a**) Final fracture surface (darker area—suffered ductile fracture, marked with white arrow). (**b**) Detail of final fracture surface with dimples—marked by white arrows. (**c**) Crack propagation region with striations marked by black arrows.

Based on the Figures 5 and 7, showing the microstructure of as-received material and the material that was previously exposed to mechanical fatigue, it can be seen that the microstructure of both specimens consists of polygonal ferrite with lamellar pearlite. This is in accordance with the chemical composition and type of steel applied in the experimental study. There are traces of rolling structure present, Figures 5b and 7b. No significant differences between the as-received material (specimen 1/unfailed, unbroken, untested material) and the material previously subjected to fatigue (specimen 3/failed, fractured

specimen due to mechanical fatigue) were observed. Considering the microstructure of the material previously subjected to creep, Figure 6, signs of material degradation were observed in comparison with the as- received material, Figure 5a. Namely, lamellar pearlite of the as-received material is replaced by spheroidal pearlite. In addition, Figure 6a, there are voids that occur predominantly at grain boundaries (white arrow) and non-metallic inclusions (black arrow) (Figure 6a,b). These phenomena can be attributed to the creep process, i.e., to the exposure of the material to high temperature and stress. Figure 7 shows the fracture surface of the specimen 3 caused by mechanical fatigue of the material under the action of maximum uniaxial stress of ±210 MPa after 7,142,296 cycles at a cycle frequency of 25 Hz. Final fracture surface is shown in Figure 8a, having a relatively small darker area marked with white arrow. This area suffered a ductile fracture, revealed by the dimpled surface as in Figure 8b. Dimples, marked by with arrows, Figure 8b, indicate spots of microvoid formation around non-metallic inclusions, followed by microvoid growth and coalescence [22]. In Figure 8c crack propagation area is presented, containing the characteristic striations (black arrows), that represent the incremental growth of the fatigue crack. It can be seen that individual cracks propagate in different directions, forming the multi-faceted fatigue crack propagation surface. This type of fatigue surface is found in polycrystalline materials having the transgranular fracture, with the most active slip planes dictating the fracture itself [23]. To be intrduced with the fatigue, tensile as well as some other properties of another material such as Al Si Cu alloys it is recommended to have an insight in [24]

## 4. Conclusions

The paper presents and analyzes experimentally obtained data related to mechanical properties, creep behavior and fatigue of S235JRC + C material. Experimentally obtained data can be useful in the process of designing appropriate structures. The results of experimental testing can be summarized as follows:

- Mechanical behavior of material at room and high temperatures is presented in the form of engineering stress-strain diagrams and tabular with numerical values of mechanical properties.
- It was found experimentally that the value of ultimate tensile strength of steel S235JRC + C, based on this study, occurs at a temperature of 200 °C. This fact is visible from Table 2.
- Creep behavior of S235JRC steel is presented and analyzed. It was found that this steel can be treated as creep resistant only at temperature of 400 °C and applied stress not exceeding 50% of the yield stress corresponding to this temperature.
- Fatigue behavior related to fully reversed uniaxial mechanical fatigue is presented in the form of stress-life$(S - N)$ diagram.
- Fatigue (endurance) limit calculation procedure using modified staircase method is also given and fatigue limit is determined.

The numerical results obtained at room temperature and temperature of 500 °C, $(R_{s,20° / 500\ °C})$, which relate to the maximum tensile strength $(\sigma_m / \text{MPa})$ and yield strength $(\sigma_{0.2} / \text{MPa})$ of the material, and fatigue limit at stress ratio of $R = -1$ $(\sigma_{f,R} / \text{MPa})$ at room temperature, are: $R_{(s,20° / 500\ °C)} = [\sigma_m(534/204)\ \text{MPa}; \sigma_{0.2}(488/195)\ \text{MPa}]$ and $\sigma_{f,R-1/20\ °C} = 202$ MPa.

Fatigue material test results show how repeated load can cause fracture of the element at significantly lower applied stress than fracture stress corresponding to monotonic tensile applied load. Namely, the ratio between the fatigue limit and ultimate tensile strength is 0.38. Otherwise, stress-life diagram shows the number of the cycle to failure for any of applied stress level. In addition, based on the Table 2, for each elevated temperature, a factor of reduction of the strength of the material relative to its value at room temperature is visible.

**Author Contributions:** Conceptualization, J.B.; Methodology, J.B.; S.K.; G.V.; Software, S.K.; M.B., S.B.; Validation, M.M.; S.K.; M.D.; Formal Analysis, S.K.; M.M.; M.B.; Investigation, J.B.; S.B.; M.B.; M.D.; Resources, J.B.; Data Curation, J.B.; Writing—Original Draft Preparation, J.B.; Writing—Review & Editing, J.B.; S.K.; M.M.; Visualization, S.K.; G.V.; M.D.; Supervision, S.K.; M.M.; G.V.; Project Administration, J.B.; Funding Acquisition, J.B. All authors have read and agreed to the published version of the manuscript.

**Funding:** This work has been fully supported-supported in part by Croatian Science Foundation under the project IP-2019-04-8615, (2019). The authors are grateful for the support received.

**Institutional Review Board Statement:** Not applicable.

**Informed Consent Statement:** Not applicable.

**Acknowledgments:** This work has been fully supported-supported in part by the University of Rijeka within the projects uniri-technic-18–42 and uniri-technic-18–200. The authors would like to thank all the funding received.

**Conflicts of Interest:** The authors declare no conflict of interest.

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
