# Peer review of "S235JRC+C Steel Response Analysis Subjected to Uniaxial Stress Tests in the Area of High Temperatures and Material Fatigue"

_sustainability, doi:10.3390/su13105675_

Round 1
Reviewer 1 Report
- Lines 154, 155, 156, 162 etc. what it means ? It means that δm depend on the T - temperature ... please dont use all known symbols in regresion equations. Second question i think that beetter try firstly find and describe method of the aproximation than use some Microsoft Office (there are more adequate methods). You also should do error of the aproximation calculation.
- Table 2 description is incomprehensible, are there non numerical values ?? Next line 173 authors write "Based on the experimental tensile tests, Figure 2 (a, b), as well as the data given in Table
2, a trend of change of mechanical properties with increasing temperature is visible. " so this values are from experiment or from aproximation ? Generaly a some chaotic presentaion of data and experiment results. It must be corrected. - Table 3. Please rethink why there are some diferencies and write about influence of alloying elements on steel properties. And same table 4. This chapter is rather poorly writen and should be rethinked by authors.
- Lines 183-185 what phenomena causes this changes of material properties ?
- Pharagraph 3.2 the beginning of the text is a theoretical basis that should rather be found in chapter 1.
- Figures 3-5 are the same variotion and better use one fig and leters a-c to describe. And this figures i think are calculated based on experimet reserch, but how and what reserch were made is for me unclear (plaease use extended description of reserch and method presented in Youre paper).
- Same objections to the presentation concern the following chapters, for example, line 319 the authors write the modified staircase method was used - no description, no reference, etc. Again, the results are presented in a chaotic manner, which makes their reception difficult. If the description is starting in line 327 the results should be after the decription.
- The analysis of the microstructure is the strongest point of the article. However, the analysis of the temperature-sensitive area (described earlier is necessary). The authors focus on three samples, but the conditions for which they were obtained should be clarified.
- The conclusions are sometimes rather summaries and statements. They require a synthesis of the results
- Correct language, however, requiring correction, colored fonts, captions under figures and descriptions of tables, often not clear and illegible.
Author Response
Thanks for your valuable comments.

Reviewer 2 Report
This paper presents some experimental results about the mechanical properties of a structural steel S235JRC+C (1.0122).
A paper that contains experimental data of the mechanical properties of a common material is always interesting, but it is surprising the lack of this kind of information in the technical literature specially, as the authors claim, for “the most frequently used steel grade worldwide”.
I have some questions that should be clarified before proposing this paper for publication.
- The authors found that “it is also evident that the material has sufficiently high level of mechanical properties up to a temperature of 400°C, which is advantageous given the use of this material”, but the experimental creep tests were performed above this temperature (400ºC, 500ºC and 600ºC). I think that the study should be performed in the range of temperatures where the material behaves correctly. Therefore, the sentence “Experimental research, conducted in this paper, shows its weak creep resistance” must be modulated, because it is true for high temperatures but it has not been studied for temperatures lower than 400ºC.
- The polynomic expressions that fit the experimental results do not have physical meaning and in consequence, they do not give any useful information. They are not the result of applying a model that predicts the reality, they are just algebraic expressions that adjust the experiments. So, I do not understand the paragraph of lines 192-195 because there is no physical model that fits the results. The authors must explain this point or suppress the polynomic expressions and the corresponding explanation.
- I suggest representing in Figure 2 (b) the reference curve (20ºC).
- It is not necessary to write continuously throughout the text and the figures the kind of steel, it is enough when it is compared to other materials. On the contrary, I think that it should appear in the title of the paper instead of a “low-carbon steel”.
- Specimens in figure 1 should be represented in the same way: same arrows, same line thickness, etc.
- I am not sure that the data in Table 5 are correct: I see 9 experiments in Figure 6 for the three stress levels. I do not understand the three fails of specimens 3, 4 and 5. It does not affect the final result, but I would like an explanation.
- It would not be Equation (2) logarithmic?
- It is confusing to use σ for the standard deviation and μ for the mean value of stress.
- I do not see which is the interest of the microstructure analysis of the material. It does not give any further information.
- The conclusions must be rewritten. In fact, the conclusions of the paper are clearer explained in the abstract.
- I do not understand to acknowledge two of the authors of the paper in the section “Acknowledgments”.
Author Response
Thanks for your valuable comments.

Round 2
Reviewer 1 Report
1. Line 182 and others still there is T and it should be aded "where T is temperature"
Reviewer 2 Report
The authors have answered my questions. I propose this paper for publication